# Antibacterial and Antiparasitic Activity of Propyl-Propane-Thiosulfinate (PTS) and Propyl-Propane-Thiosulfonate (PTSO) from *Allium cepa* against Gilthead Sea Bream Pathogens in In Vitro and In Vivo Studies

**DOI:** 10.3390/molecules27206900

**Published:** 2022-10-14

**Authors:** Jose F. Cabello-Gómez, María Arántzazu Aguinaga-Casañas, Ana Falcón-Piñeiro, Elías González-Gragera, Raquel Márquez-Martín, María del Mar Agraso, Laura Bermúdez, Alberto Baños, Manuel Martínez-Bueno

**Affiliations:** 1Aquaculture Technology Centre of Andalusia, CTAQUA, Muelle Comercial s/n, El Puerto de Santa María, 11500 Cadiz, Spain; 2Department of Microbiology and Biotechnology, DMC Research Center, Camino de Jayena s/n, 18620 Granada, Spain; 3Department of Microbiology, University of Granada, Fuente Nueva s/n, 18071 Granada, Spain

**Keywords:** *Photobacterium damselae*, *Sparicotyle chrysophrii*, antibacterial, antiparasitic, *Allium*, phytogenics

## Abstract

The use of phytogenic extracts is considered a sustainable strategy for the prevention of fish diseases, including *Alliaceae* as a potential option due to their variety of bioactive compounds. In this study, we analyzed the antibacterial and antiparasitic potential of propyl-propane-thiosulfinate (PTS) and propyl-propane-thiosulfonate (PTSO) from onions. The in vitro activity against *Pseudomonas anguilliseptica*, *Tenacibaculum maritimum*, and *Photobacterium damselae* of both compounds was tested. In addition, the viability of *Sparicotyle chrysophrii* larvae was evaluated. Moreover, a diet that consisted of a blend of PTS/PTSO (ALLIUM) was used. A total of 90 gilthead sea bream juveniles were tested against *P. damselae* subsp. *Piscicida* after 12 weeks of dietary administration. Furthermore, 150 fish with a rate of 10–15 parasites/fish were fed for 21 days and the number of gill parasites was recorded. All strains were sensitive to both compounds. PTSO showed the highest inhibitory effect against all target strains, while PTS showed higher effectiveness against *S. chrysophrii.* Fish from ALLIUM group presented the highest probability of survival, increasing up to 91.1%, whereas in the control group, the probability of survival was 66.7%. The number of parasites in the gilthead sea bream decreased in the ALLIUM group over time. These results suggest the inclusion of PTS and PTSO in feed as a natural strategy to prevent antibacterial and antiparasitic fish diseases.

## 1. Introduction

The implementation of intensive and semi-intensive production methods, coupled with the diversity of farmed species and the decline in ocean fish stocks, has positioned aquaculture as the fastest-growing food production sector [1,2]. Among the different types of marine fish production, the gilthead sea bream (*Sparus aurata*) is one of the most important farmed species in the Mediterranean, increasing steadily from around 96,718 tons in 2004 to an estimated 258,753 tons in 2019 and valued at USD 1.28 billion worldwide [3]. However, this rapid development in marine cages and the living conditions in aquaculture systems is associated with the appearance of diseases that can have significant economic impacts on the industry.

One of the most important bacterial infections is photobacteriosis/pasteurellosis, caused by *Photobacterium damselae* subsp. *Piscicida*, a Gram-negative bacterium that causes significant production losses due to its ability to infect a wide range of fish species with high mortality rates from lesions, hematological disorders, and histological abnormalities, among other clinical signs [4,5]. Gilthead sea bream can also be affected by parasites such as *Sparicotyle chrysophrii*, a common monogenean parasite that is often responsible for large commercial losses in fish farming and causes lethal diseases through anemia, histopathological damage, hemorrhages and necrosis [6,7]. These and other infections with pathogenic bacteria and parasites that can cause veterinary problems are common in farmed fish, so the use of preventive or therapeutic agents is necessary to avoid the appearance and spread of diseases. However, the growing restriction on the use of drugs recommended by the FAO (Food and Agriculture Organization of the United Nations), OIE (World Organisation for Animal Health) and WHO (World Health Organization) [8] highlights the importance of being aware of the usage patterns and their impact on public health. This pressure has pushed the sector to look for innovative strategies to prevent and minimize the risk of diseases.

To continue with further sustainable development of the aquaculture industry, several alternative biocontrol methods have been proposed, including good management practices, such as reducing animal density, vaccinations, and phage therapy, together with nutritional interventions that comprise the use of functional additivess, such as probiotics, prebiotics, symbiotics, and phytogenics [9,10,11,12].

Plants contain numerous secondary metabolites that play functional roles, such as protecting plants from herbivores, insects, microbial infections and other challenges, with antimicrobial, anti-inflammatory, antioxidant, and antiparasitic properties, the reason for which their use as traditional medicinal plants has widely expanded [13]. These bioactive compounds can be isolated from many different natural sources, including roots, herbs, and bulbs, that are readily available for use as components of animal feed [14,15,16].

Among the different phytogenic plant extracts, those derived from *Allium* spp., such as garlic (*Allium sativum*) and onion (*Allium cepa*), hold great potential due to their variety of bioactive compounds, including polyphenols and organosulfur compounds, among others, being one of the most studied plants of therapeutic importance [17]. In aquaculture, sulfur-containing compounds derived from *Alliaceae* are known to exhibit antioxidant, antimicrobial, antifungal, and antiparasitic properties, and have also been reported to modulate immune response and upgrade mucosal epithelial barrier functions [18]. The most frequent organosulfur compounds present in onion are isoalliin (S-propenyl-L-cysteine sulfoxide); methiin (S-methyl-L-cysteine sulfoxide); and propiin (S-propyl-L-cysteine sulfoxide). The latter, due to the action of alliinase, is transformed into propyl-propane-thiosulfinate (PTS), which, through dismutation or disproportionation reactions, leads to dipropyl disulfide and propyl-propane thiosulfonate (PTSO) [19]. Both organosulfur compounds (Figure 1) stand out due to their stability that means they are suitable for use in feed processes [20], together with multiple functional properties that may contribute to reducing the risk of diseases in fish farming. These compounds have shown significant antimicrobial activity against Gram-positive and Gram-negative multidrug-resistant bacteria isolated from human samples [21]. Moreover, the use of PTS-PTSO compounds has also shown an improvement in epithelial mucosa in colitis mouse models [22]. In terrestrial animals, the addition of PTS/PTSO as a feed additive has been demonstrated to improve animal health status through modulating intestinal microbiota in the most relevant livestock species, such as broiler chickens, laying hens, and growing and weaned piglets [23,24,25,26,27]. Furthermore, PTSO has shown antiparasitic activity against *Eimeria acervulina* in broilers [28]. Although the health benefits of these compounds have been widely described for livestock species, the literature on aquatic animals is limited.

In previous studies, we have reported the antiparasitic effects of PTSO against salmon sea lice (*Caligus rogercresseyi*) and fish nematodes, such as *Anisakis* and *Hysterotylacium aduncum* [29]. Nevertheless, there are no in vivo trials that show the potential of these organosulfur compounds against other pathogens of aquatic interest, such as *Sparicotyle chrysophrii*, a gilt parasite that belongs to the monogenean class, specific to *S. aurata*, causes high mortality with several financial impacts and whose prevalence and virulence is increasing in warmer seawater due to climate change [6,30].

Considering this background information, in the present study, the in vitro antibacterial and antiparasitic effects of PTS and PTSO were studied. Moreover, the effect of both compounds in juvenile gilthead sea bream (*S. aurata*) studied with *P. damselae* and *S. chrysophrii* was also evaluated.

## 2. Results

### 2.1. In Vitro Antibacterial Effect

The susceptibility of the bacterial pathogens responsible for the main diseases in *S. aurata* (Table 1) to organosulfur compounds PTS and PTSO was tested. The obtained disk diffusion data set was highly diverse, as it encompasses a wide range of inhibition zones. However, despite the diversity, all pathogenic strains proved to be sensitive to both compounds. PTSO exhibited a high inhibitory effect (≥20 mm) against all target strains from 10 mg/mL upwards, whereas PTS had such an effect from 25 mg/mL upwards. In all cases, the diameters of inhibition zones were directly related to the concentration of antimicrobials. Nevertheless, in the case of PTS against *Photobacterium damselae* subsp. *damselae*, no significant increase in the inhibition zone was observed as the concentration increased. This is also displayed in the figure below (Figure 2).

More precise data on the in vitro antimicrobial activity of PTS and PTSO were obtained through the determination of the minimum bactericidal concentrations recorded (Table 2). The positive control displayed the expected antimicrobial activity against the different bacterial strains at the tested concentration. On the other hand, the lowest MBC value registered corresponds to PTSO against *P. anguilliseptica* and *P. damselae* subsp. *damselae,* displaying an MBC of 39.06 and 78.125 µg/mL, respectively, whereas *T. maritimum* turned out to be the most resistant strain, since the MBC values of PTS and PTSO were 2500 µg/mL and 1250 µg/mL, respectively.

### 2.2. In Vitro Antiparasitic Effect

No significant differences were observed between replicates, so data from the same treatment were pooled. As expected, the use of 4-hexylresorcinol as a positive control killed all the parasites from the first measure. Larvae exposed to PTS showed a lower probability of survival than the control group from 0.5 µg/mL upwards (*p* < 0.001), while those exposed to PTSO did so from a compound concentration of 1 µg/mL upwards (*p* < 0.01) (Figure 3). At the end of the test (60 min), the survival probability of parasites incubated at 0.5, 1 and 5 µg/mL of PTS was reduced by 43.3%, 56.7% and 90.0%, respectively. On the other hand, exposure to PTSO at the same concentrations reduced the survival probability of the larvae by 13.3%, 33.3% and 50%.

### 2.3. In Vivo Survival Effect against Photobacterium *subsp*. Piscicida

The fish fed with PTS/PTSO presented a higher probability of survival (*p* < 0.01) compared with those fed the basal diet (Figure 4). During the development of the experiment, both groups presented the highest mortality between days 4 and 9, stabilizing from day 10 until the end of the trial and showing a final survival probability of 66.7% in the control group and 91.1% in the ALLIUM group.

### 2.4. In Vivo Antiparasitic Effect

Although the initial infestation level was similar in both groups during the first week, an increase in parasites per fish was observed in the control group, whilst a significant reduction was observed in the group of fish fed with *Allium* compounds. This difference increased over time, resulting in a mean infestation value of 28.1 ± 3.33 parasites per fish in the control group in the last week, compared to 4.1 ± 2.40 in the experimental group (*p* < 0.001) (Figure 5).

## 3. Discussion

The antimicrobial potential of the phytogenic molecules present in medicinal plants has been widely described in the literature, especially those compounds derived from *Alliaceae* [31,32]. Indeed, it is thought that their bioactive components may be able to reduce the use of antibiotics and replace them as promising additives for use in fish farms [33,34]. This study evaluated the in vitro and in vivo antimicrobial activity of two allium-derived organosulfur compounds, PTS and PTSO, against bacteria and parasites of concern in gilthead sea bream farming.

Our results showed that PTS and PTSO were able to inhibit *P. anguilliseptica*, *T. maritimum* and *P. damselae*. These results were similar to those obtained by Guo et al., who reported antibacterial activity of garlic powder against *P. damselae* subsp. *piscicida*, although obtaining lower inhibition zones and higher MCB values than those found in our in vitro tests with PTSO [35]. Similarly, and in addition to *Alliaceae*, other bioactive compounds have been shown to exert activity against these bacteria. Isothiocyanates from horseradish (*Armoracia rusticana*) were also able to inhibit *P. anguilliseptica*, *T. maritimum* and *P. damselae,* among others, although, once more, obtaining smaller inhibition halos compared to PTSO [36]. Nevertheless, comparing both studies, PTS showed similar antibacterial potential. In addition, the highest concentration of aqueous extract of *Origanum vulgare* also exerted antibacterial activity against *P. damselae* [37].

It is assumed that the main antibiotic mechanism of action of organosulfur compounds is, together with its high permeability through phospholipid membranes, their interaction with certain thiol-containing enzymes in the microorganism. This reaction causes the inhibition of thiol-dependent enzymatic systems that could explain its activity against a wide range of Gram-negative and Gram-positive bacteria [38,39]. Considering the chemical structure of PTS and PTSO, this could also be applicable to these compounds and could explain the results obtained in our study. According to this result, PTS and PTSO were reported to exert a significant broad spectrum of antibacterial activity against several bacterial groups of human and livestock of interest, including a selection of Gram-negative and Gram-positive multi-resistant bacteria isolated from human clinical samples [21,40,41].

The reason for which PTSO showed higher antibacterial potential than other similar compounds could be related to its stability. The major phytochemicals derived from garlic (*Allium sativum*) that exhibit antibacterial activity are oil-soluble organosulfur compounds, including allicin and allyl sulfides, among others [42]. Nevertheless, allicin is easily degraded under the influence of temperature to ajoenes and vinyldithiins, which are usually isolated from different kind of garlic extracts [43]. Moreover, due to its instability, its antibacterial activity is quickly reduced in less than one hour in the presence of vegetal oil [44]. In onions, PTS is one of the most common sulfur bioactive compounds and, unlike allicin, it is more stable. Even so, through dismutation or disproportionation reactions, PTS changes into dipropyl disulfide and PTSO is the most stable [45].

The origin of *Alliaceae* bulbs, their processing before obtaining extracts, as well as the extraction method used, have a great influence on the concentration and type of bioactive compounds, and consequently on the results obtained when their antibacterial ability is evaluated [46]. Therefore, for the inclusion of these compounds in aquafeeds, it is of high importance to guarantee the standardization and traceability of the active ingredients once they are added to the feed. In this sense, it must be considered that, as mentioned before, allicin is more unstable than PTS and especially PTSO [47]. It has been shown that both compounds are detected in fish diets after their inclusion, which makes them suitable compounds to be used as feed additives [20].

In the challenge trial with *P. damselae* subsp. *piscicida*, the inclusion of PTS/PTSO in the diet provided a preventive effect that was reflected in the increased survival rates. Our results are in accordance with those reported by other authors that obtained lower mortality rates in fish that were fed with garlic and onion extracts and studied with these bacteria. The addition of 1% garlic powder to the diet of cobia fish (*Rachycentron canadum*) increased the relative percentage of survival of infected fish up to 51.6% [35]. Likewise, mugil larvae that received 100 mg garlic extract/Kg feed showed significantly increased survival rates after being challenged with *P. damselae* [48]. Other studies in which fish were challenged with other Gram-negative bacteria, such as *Pseudomonas fluorescens,* presented lower mortality rates in groups that received 1 and 3% garlic extract in their diet [49]. This increased survival in fish fed with garlic and onion extracts has also been observed in unchallenged fish, in which other aspects, such as feed utilization and blood parameters, were also improved [50,51,52].

The intestinal mucosa is the main barrier against the entry of pathogens, not only in fish but also in other species. Its protective function is conditioned by several factors, such as immunity and the composition of the intestinal microbiota. [53,54]. The health-promoting properties of *Alliaceae*-derived organosulfur compounds in different fish species and their relationship to mucosal immunity are well described. In summary, organosulfur compounds can induce immune responses and anti-inflammatory counterattacks and enhance the induced immunity by activating the TRPA1 and TRPV1 channels of enteroendocrine cells [18,55]. This mechanism of action could have allowed the preventive effect of PTS and PTSO observed in this study, preparing the defense of the intestinal mucosa before the challenge. Although further studies are needed to analyze their effect on the repair of intestinal damage, this hypothesis could be supported by the fact that PTSO was able to reduce the production of proinflammatory cytokines and improve the histological structure of the mucosa in murine models of acute and chronic colitis induced by DNBS and DBS and also in mice, induced by enteric parasites. [22,56]. Moreover, PTSO has also shown its ability to modulate intestinal microbiota of gilthead sea breams juveniles by decreasing the relative abundance of potentially pathogenic *Vibrio* and *Pseudomonas* in the foregut and hindgut, whilst increasing beneficial *Lactobacillus* [57]. These modulation effects has also been observed in other non-aquatic species, such as pigs or laying hens [23,26,27].

In our study, we also evaluated for the first time, both in vitro and in vivo, the effects of PTS and PTSO against *S. chrysophrii*. The in vitro results showed a dose-dependent antiparasitic effect of both compounds, suggesting that PTS has the highest antiparasitic potential. Nonetheless, our results are similar to those reported in other in vitro and in vivo studies, in which the potential of *Allium* extracts against other monogenean gill parasites was analyzed. The in vitro exposure to garlic aqueous extract of the monogenean *Gyrodactylus turnbulli* caused the cessation of parasite movement, showing a positive correlation between garlic concentration and time to detachment and death. Moreover, guppy fish (*Poecilia reticulata*) fed with garlic-supplemented diets showed reduced mean prevalence and mean intensity of parasites [58].

Doan et al. reviewed the efficacy of different phytogenic compounds, including those derived from garlic, against different gill parasites, concluding that the observed antiparasitic capacity of these molecules could be due to the positive immunological effects of allicin in fish, resulting in improved fish immune responses against parasites [59]. Similarly, other authors that summarized the antiparasitic effects of different plant compounds found that, in the case of *Allium* compounds, z-ajoene was the molecule that presented the highest antiparasitic activity and hypothesized that it could be due to the capability of this molecule to modify membrane proteins and lipid trafficking and enhance the expression of Th2 markers in the host cell that indicated a targeted immune response against parasites [18,60]. Indeed, the impact of *Allium* compounds against fish parasites such as *Gyrodactylus turnbulli* was reported to produce similar results to those obtained when using the antiparasitic agent levamisole [61]. In addition, another study, in which different *Allium* compounds were tested against *Spironucleus vortens*, showed that dithiins presented the lowest MIC value against this parasite, followed by ajoene and allicin [62]. Moreover, garlic has also been shown to be effective against *S. chrysophrii* when combined with other essential oils (EOs), such as carvacrol and thymol, and when included in the diet in a microencapsulated form, it reduces up to 78% of total infestation [63]. These authors also analyzed gene expression during the infestation process and they found that the transcriptomic analyses of gills of fish fed the EOs diet showed an up-regulation of genes related to tissue-specific pro-inflammatory immune response, mediated by degranulating acidophilic granulocytes and supported by anti-inflammatory and antioxidant responses, suggesting that dietary EO application may be used as a preventive and active treatment for this particular ectoparasite. By contrast, Mladineo et al. found that although natural compounds presented antiparasitic action, they were less toxic than synthetic compounds against adult *S. chrysophrii* [64]. Among the different phytogenic compounds tested, curcumin and cedrol presented the lowest LC_50_ values, garlicin and eucalyptol showed intermediate effects, while r-camphor was the least effective. Nevertheless, another in vitro study demonstrated that curcumin was not effective against the parasite *Neobenedenia girellae* [65]. These studies revealed that the use of compounds derived from *Allium*, to a greater or lesser extent, is effective for the control of gill parasites, although their degree of efficacy will depend, among other factors, on the species of the target parasite, the composition of the active ingredients and the dose of inclusion in the feed.

## 4. Materials and Methods

### 4.1. Allium Compounds

PTS and PTSO (95% purity) were supplied by DOMCA SAU (Granada, Spain) and dissolved in polysorbate 80 at 20%.

### 4.2. Bacterial Strains and Growth Media

The target strains used in this study were obtained from the Spanish Collection of Type Cultures (CECT) and the German Collection of Microorganisms and Cell Cultures (DSMZ) (Table 3), and they were stored at −70 °C with 20% glycerol. Recommendations of the Spanish and German Collections were followed concerning solid culture media and growth conditions. Bacterial suspensions were prepared in saline solution and tryptic soy broth (TSB), supplied by Scharlau (Barcelona, Spain), was used as the liquid culture media [66].

### 4.3. Parasites

*S. chrysophrii* parasites were obtained from donor fish provided by an offshore private fish farm, located in the Mediterranean Sea. Fish were placed in seawater containers with oxygen supplementation, and transported to the facilities of the Aquaculture Technology Centre of Andalusia, CTAQUA (El Puerto de Santa María, Spain). Following this, 20 donor fish with an average weight of 120 g and a level of infestation of 1 parasite/fish in the first-gill arc were distributed in 2400 L truncated cone tanks provided with drains, with a layer of synthetic nylon fiber as a filter to retain the floating eggs released by the parasitized gilthead breams. This technique allowed us, after the hatching of eggs and in the presence of host fish, to reproduce the complete life cycle of the parasite.

### 4.4. In Vitro Tests

#### 4.4.1. In Vitro Antibacterial Activity

Two well-established procedures were carried out to assess the antibacterial activity of PTS and PTSO. The susceptibility of the target strains to both organosulfur compounds was screened by the disk-diffusion method proposed by Kirby–Bauer [67] and modified by Calvo and Asensio [68]. Firstly, sterilized cellulose discs of 6 mm in diameter (Whatman^®^ antibiotic test discs, Buckinghamshire, UK) were impregnated with 20 µL of PTS or PTSO at a dose of 2.5, 5, 10, 25, and 50 mg/mL, and placed in the center of agar plates that were previously inoculated with the target strains. To achieve confluent growth after incubation, the bacterial suspensions used to inoculate the agar plates were adjusted to 1 × 10^6^ CFU/mL. Three independent tests were carried out, in which each sample was tested in duplicate. The averages of all determinations were categorized as follows (disk diameter included): inhibition zones ≥ 20 mm were considered as strongly inhibitory, zones from 12 to 20 mm were considered as moderately inhibitory, and <12 mm were weak or not inhibitory [69,70]. The EC50 values were calculated.

The Minimum Bactericidal Concentration (MBC) of PTS and PTSO was determined using the standard broth microdilution method, according to the Clinical and Laboratory Standards Institute (CLSI) guidelines [66]. Briefly, bacterial suspensions were diluted to obtain a concentration of 1 × 10^5^ CFU/mL. Then, 1:2 dilutions of both PTS and PTSO were prepared in 96-well microtiter plates to achieve the final concentrations, ranging from 4.88 to 10,000 μg/mL, and inoculated with 30 μL of the corresponding bacterial suspension. Each plate included a positive and negative control, without the tested compound or neither compound nor bacteria, respectively. Moreover, wells with a concentration of ceftazidime (8 µg/mL) were used as a second positive control. Absorbance was measured at 620 nm and cell growth was tested by culturing in agar plates. The lowest concentration of PTS/PTSO at which no growth was observed was defined as the MBC. MBC determination was performed in triplicate.

#### 4.4.2. In Vitro Tests for the Evaluation of the Antiparasitic Activity

*S. chrysphriies* larvae were obtained from fragments of nylon fiber inside the tanks of donor gilthead sea breams infected with the parasite. Fibers were continuously kept hydrated with salt water from the same tanks, until the evaluation of the antiparasitic activity. The larval stages of the parasite among the fibers were located by using a binocular loupe (20/40 magnification) and transferred to a 12-well culture plate (ThermoScientific, Roskilde, Denmark), placing 10 larvae in each well. Parasites were exposed to PTS and PTSO at 0.5, 1 and 5 µg/mL and a control treatment with salt water was used. Additionally, a positive control with 4-hexylresorcinol at 1 µM was used to compare and detect dead parasites. All treatments were performed in triplicate. Viability was ensured at 0, 5, 30 and 60 min, considering the absence of mobility as a mortality indicator.

### 4.5. In Vivo Challenges

#### 4.5.1. Fish and Experimental Conditions

Juvenile gilthead sea breams (*S. aurata*) were allocated in 400 L tanks with a recirculating RAS, equipped with physical and biological filters. The temperature was maintained at 21 ± 1 °C with a photoperiod regime of 12:12 h (light:dark) and daily water renewal of 10%. Water temperature (Oxyguard^®^ Handy Polaris probe, Farum, Denmark), oxygen (Oxyguard^®^ Handy Polaris probe, Farum, Denmark) and pH (Oxyguard^®^ Handy pH probe, Farum, Denmark) were controlled daily. Salinity (Aquafauna Bio Marine^®^ ABMTC refractometer, Hawthorne, CA, USA), ammonia (Merck MQuant^®^ Ammonium Test, Darmstadt, Germany) and nitrite (Merck MQuant^®^ Nitrite Test, Darmstadt, Germany) were monitored weekly. Fish were kept in these tanks until their use in antibacterial or antiparasitic in vivo challenge experiments. All animals were handled following the European Union Guidelines (Directive 2010/63/UE) for the use of laboratory animals.

#### 4.5.2. Diets

The experimental diets followed a standard and commercially available basal formula for gilthead sea bream, containing fish meal, poultry by-product meal, oil seed, vitamin–mineral premix and soybean and rapeseed oil, among other ingredients (NUTRAPLUS, Dibaq, Segovia, Spain). PTSO and PTS were added to the basal formula in a proportion 1:1; *w*/*w* (150 mg/kg). Once the meal was homogenized, the granulated fish feed was manufactured by SPAROS (Olhão, Portugal). Table 4 shows the nutritional composition of the experimental diet. A diet without additives was prepared as a control. Finally, to ensure the concentration of PTS/PTSO in the diets, UHPLC-ESI-MS/MS analyses were performed, according to the method described by Abad et al. [71].

#### 4.5.3. Resistance to Experimental Infections against *Photobacterium damselae* subsp. *piscicida*

For this trial, juvenile gilthead seabreams (*n* = 90) were placed into 85 L tanks with the same initial biomass each (15 fish per tank, initial body weight of 37 ± 1 g, load of 6.5 kg/m^3^) and randomly assigned to two experimental groups in triplicate (3 × 15; *n* = 45 each). The control group was fed the non-supplemented diet during the whole experiment, while the ALLIUM group was fed a diet supplemented with 150 mg/kg of PTS/PTSO for 12 weeks prior to the start of the fish health challenge and afterwards, they were fed with the non-supplemented diet during the challenge. For infestation, the fish were previously anesthetized with 80 mg/L of tricaine methane sulfonate (MS-222) and were subjected to intraperitoneal injection with 100 µL of *P. damselae* subsp. *piscicida* DSM 22834 suspended in phosphate buffered saline (PBS), at a dose of 5 × 10^6^ CFU/fish. The bacterial concentration in the inoculum was determined by ten-fold serial dilutions and further plating onto Columbia blood agar (Scharlau). The concentration of the bacterial suspension corresponds to the value of lethal dose 50 (LD50), which was previously determined. The control group of fish was inoculated with 100 µL of sterile PBS as a negative control.

Once the fish were infected, they were monitored for 21 days. The fish were observed twice a day, monitoring their state of health, with a special interest in those individuals that showed photobacteriosis symptomatology, and in particular, the mortalities that may appear. Moribund fish were humanly sacrificed for dissection and observation of internal organs. During the whole fish health challenge period, daily mortality was recorded.

#### 4.5.4. Resistance to Experimental Infections against *Sparicotyle chrysophrii*

To obtain a similar level of infestation, 150 healthy fish (average weight of 90 ± 0.1 g) were placed in tanks for 30 days with pieces of nylon fibers that contained parasite eggs. When the fish presented a homogeneous rate of infestation (around 10–15 parasites/fish), they were transferred to 6 tanks of 300 L at a ratio of 20 fish per tank (initial body weight 114 ± 0.5 g, load 7.6 Kg/m^3^). Fish were assigned to two dietary treatments (3 tanks per treatment), consisting of the non-supplemented diet for the control group and a diet that contained 150 mg/kg of a blend of PTSO and PTS in proportion 1:1; *w*/*w* for the ALLIUM group, over three weeks. Each week, 5 fish from each tank were euthanized with MS-222 (Sigma Aldrich, Madrid, Spain) at a dose of 200 mg/L and their gills were removed and placed on Petri dishes with salt water. Then, they were examined under a binocular loupe by combing the slides with the help of a needle, recording the number of parasites in each bronchial arch.

### 4.6. Statistical Analyses

Data were statistically analyzed by using the Real Statistics Resource Pack software (Release 7.6) (www.real-statistics.com, accessed on 6 June 2022) [72]. The effects of antimicrobial activity, analyzed using the inhibition halos, were expressed as the mean ± standard deviation (SD). Survival curves for *Sparicotyle* in the in vitro test and *Photobacterium* in the in vivo challenge were estimated by the Kaplan–Meier method and compared to controls by the log-rank test. The differences in the number of parasites per gill in *Sparicotyle* in the in vivo tests were compared using Student’s tests (using treatment as the fixed effect at 0, 7, 14, and 21 days). The statistical significance level was accepted at *p* < 0.05.

## 5. Conclusions

In this study, the capacity of the organosulfur compounds PTS and PTSO from *Alliaceae* extracts against common pathogens in aquaculture was analyzed. Our results show that both phytogenics have antibacterial and antiparasitic activity. Although further studies including other stages and farm conditions should be performed, the increase in survival probability against *P. damselae* subsp. *Piscicida* and the reduction in the gill parasites *S. chrysophrii* of gilthead seabreams suggest the use of these *allium*-derived compounds as a natural diet strategy to prevent fish diseases in aquaculture.

## Figures and Tables

**Figure 1 molecules-27-06900-f001:**
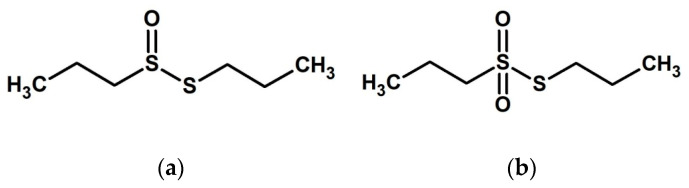
Chemical structure of PTS (**a**) and PTSO (**b**).

**Figure 2 molecules-27-06900-f002:**
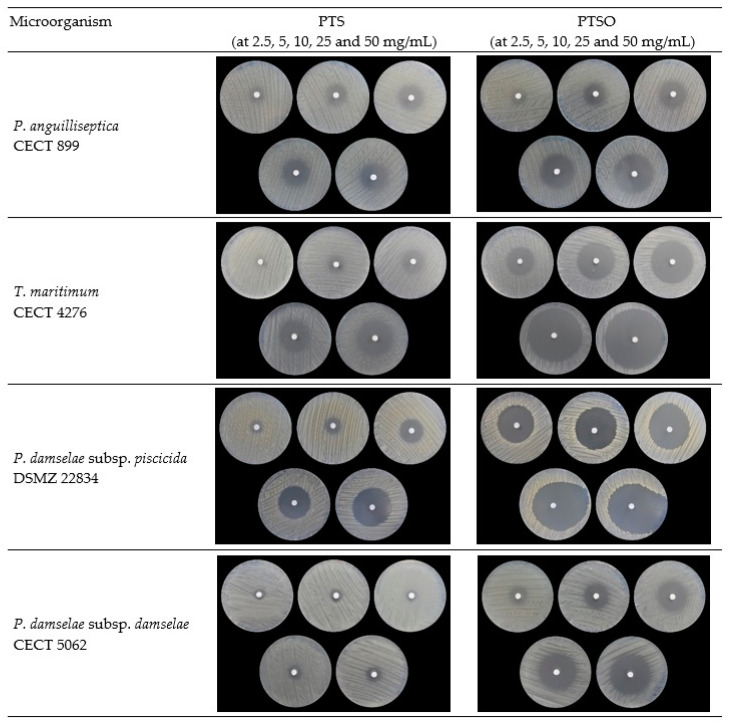
Results of the antibiosis test in solid media of PTS and PTSO at different concentrations against the target strains.

**Figure 3 molecules-27-06900-f003:**
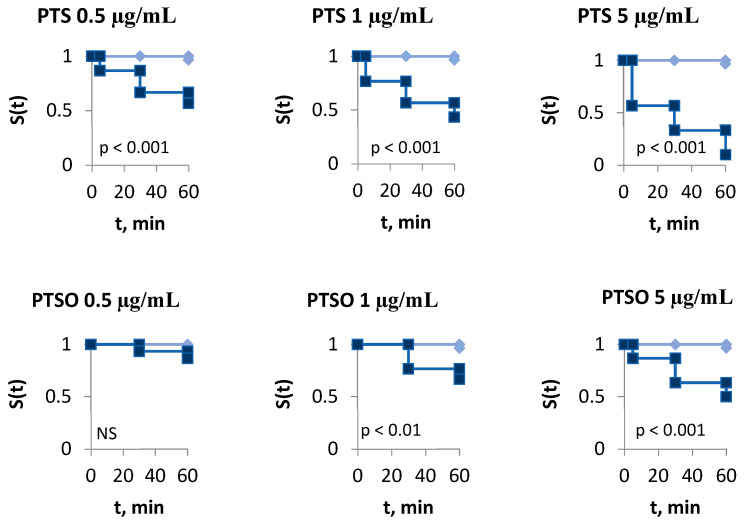
Kaplan–Meier survival curves for *Sparicotyle chrysophrii* larvae incubated at different concentrations of PTS and PTSO (0.5, 1 and 5 µg/mL each) at 0, 5, 30 and 60 min compared to control, using the long-rank test at a 95% confidence level. NS, not significant. ♦: Control group; ■: treatment group.

**Figure 4 molecules-27-06900-f004:**
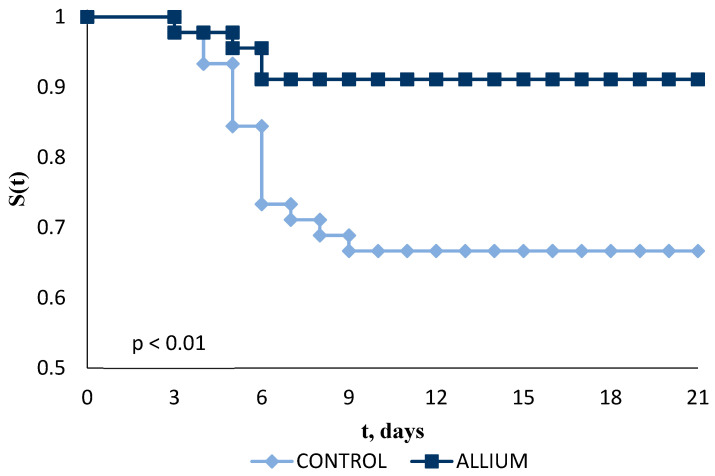
Kaplan–Meier Survival curves for seabream (*S. aurata*) fed a basal diet (control) or a diet supplemented with 150 mg/kg PTS/PTSO (ALLIUM) during 21 days of challenge test, compared to control using the long-rank test at a 95% confidence level.

**Figure 5 molecules-27-06900-f005:**
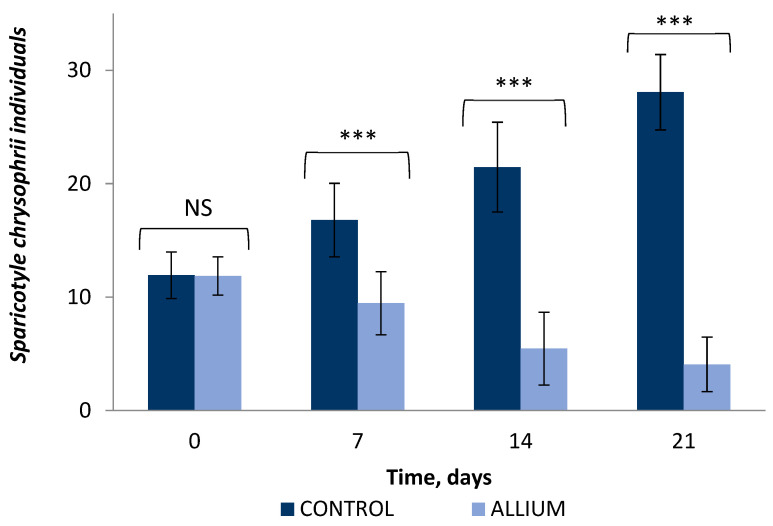
Evolution over time of the number of *Sparicotyle chrysophrii* parasites in sea bream gills in the control groups and those treated with a diet with 150 mg/kg PTS/PTSO (ALLIUM). The bars represent means ± standard deviations of three independent tests. *** *p* < 0.001; NS, non-significant.

**Table 1 molecules-27-06900-t001:** Antibacterial activity of PTS and PTSO at different concentrations against pathogenic bacteria that infected *S. aurata* by the disk-diffusion method, expressed as the average diameter ± standard deviation of the inhibition zone (mm), along with EC50 values.

Compound	Concentration (mg/mL)	*P. anguilliseptica*	*T. maritimum*	*P. damselae* subsp. *Piscicida*	*P. damselae* subsp. *damselae*
**PTS**	**2.5**	12.5 ± 1.12	0.0 ± 0.00	0.0 ± 0.00	11.3 ± 0.83
	**5**	17.5 ± 1.80	13.0 ± 1.41	11.0 ± 1.41	12.8 ± 1.09
	**10**	19.5 ± 1.80	19.0 ± 2.24	26.3 ± 2.38	13.5 ± 1.12
	**25**	23.5 ± 1.12	30.5 ± 1.80	35.0 ± 1.41	15.0 ± 0.71
	**50**	29.8 ± 2.86	34.5 ± 1.12	43.8 ± 2.38	16.5 ± 1.80
	**EC50 ^1^** **(mg/mL)**	12.7563	8.1651	8.9399	11.6285
**PTSO**	**2.5**	14.5 ± 1.12	28.5 ± 1.12	40.5 ± 1.66	18.8 ± 2.86
	**5**	17.5 ± 1.80	39.0 ± 1.41	47.5 ± 1.80	24.5 ± 1.12
	**10**	23.5 ± 1.80	45.5 ± 1.80	56.8 ± 2.86	30.5 ± 1.12
	**25**	32.5 ± 1.12	59.0 ± 1.41	62.5 ± 2.96	39.0 ± 1.00
	**50**	40.5 ± 1.12	66.5 ± 1.12	68.0 ± 2.24	43.5 ± 1.12
	**EC50 ^1^** **(mg/mL)**	14.4796	10.5698	9.0014	10.4522

^1^ EC50: Half maximal effective concentration.

**Table 2 molecules-27-06900-t002:** Minimum Bactericidal Concentration (MBC) of PTS and PTSO against target bacteria.

Strain	PTS (µg/mL)	PTSO (µg/mL)
*Pseudomonas anguilliseptica*	312.5	39.06
*Tenacibaculum maritimum*	2500	1250
*Photobacterium damselae* subsp. *piscicida*	625	156.25
*Photobacterium damselae* subsp. *damselae*	625	78.125

**Table 3 molecules-27-06900-t003:** Strains used, along with their references and isolation source.

Strain	Reference	Isolated
*Pseudomonas anguilliseptica*	CECT 899	*Anguilla japonica*
*Tenacibaculum maritimum*	CECT 4276	Kidney of diseased black seabream
*Photobacterium damselae* subsp. *piscicida*	DSM 22834	Seriola with pseudotuberculosis
*Photobacterium damselae* subsp. *damselae*	CECT 5062	Diseased turbot

**Table 4 molecules-27-06900-t004:** Diet composition of the fish experimental diet.

Nutrient Composition	
Crude protein%	45.00
Crude fat%	20.00
Cellulose%	2.8
Ash%	8.00
N.F.E.%	13.3
Moisture%	10.00
Calcium%	1.8
Phosphorus%	0.9
Gross energy (MJ/kg)	21.10
Digestible energy (MJ/kg)	17.5
Protein digestibility%	90.00
PD/DE (gr/MJ)	23.14
PTS/PTSO (mg/kg)	150

## Data Availability

The data presented in this study are available upon request from the corresponding author.

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
