# Peer review of "Antibacterial and Antiparasitic Activity of Propyl-Propane-Thiosulfinate (PTS) and Propyl-Propane-Thiosulfonate (PTSO) from Allium cepa against Gilthead Sea Bream Pathogens in In Vitro and In Vivo Studies"

_molecules, 2022, doi:10.3390/molecules27206900_

Round 1

Reviewer 1 Report

1The manuscript is well written. However, following points are to be addressed:

1.     Page 2 lines “In onion, the most common sulfur compounds are isoalliin (S-propenyl L-83 cysteine sulfoxide); methiin (S-methyl-L-cysteine sulfoxide and propiin (S-propyl-L-84 cysteine sulfoxide) that, due to the action of alliinase, leads to dipropyl thiosulfinate (PTS) 85 which is transformed into dipropyl disulfide and propyl-propane thiosulfonate (PTSO) 86 through dismutation or disproportionation reactions”     shall be modified to remove plaigraism.

2.  Page 3 line 98 Authors can go through the following references and cite them

a) https://patentscope.wipo.int/search/en/detail.jsf?docId=WO2015049587, there is a patent filed for prevention and reduction of parasites in aquatic animals.

b) 10.1016/J.EXPPARA.2010.10.001

3.       In vitro tests for the evaluation of antiparasitic activity didn’t show any positive control used.

4.       The results may be compared with some known natural compounds or synthetic ones.

Author Response

The manuscript is well written. However, following points are to be addressed:

  1. Page 2 lines “In onion, the most common sulfur compounds are isoalliin (S-propenyl L-83 cysteine sulfoxide); methiin (S-methyl-L-cysteine sulfoxide and propiin (S-propyl-L-84 cysteine sulfoxide) that, due to the action of alliinase, leads to dipropyl thiosulfinate (PTS) 85 which is transformed into dipropyl disulfide and propyl-propane thiosulfonate (PTSO) 86 through dismutation or disproportionation reactions” shall be modified to remove plagiarism.

Thank you for this observation. As this reference belongs to a paper of our research group, we included the idea while writing but we forgot to modify this part of the text.  It has been modified in the manuscript.

  1. Page 3 line 98 Authors can go through the following references and cite them
  2. a) https://patentscope.wipo.int/search/en/detail.jsf?docId=WO2015049587, there is a patent filed for prevention and reduction of parasites in aquatic animals.
  3. b) 10.1016/J.EXPPARA.2010.10.001

This patent belongs to our research group and it is cited as reference 29. Respect the second reference you mention,  Millet et al. (2011), we have followed your suggestion and it has been included as reference 62 in the discussion (lines 282-284). Thank you very much for your indications.

  1. In vitro tests for the evaluation of antiparasitic activity didn’t show any positive control used.

You are right. Although following the scientific method, we used a positive control (4-hexylresorcinol at 1µM), we did not include these results in the text as we did not want to establish a comparison with other products, just to check if the compounds have antiparasitic potential compared to the negative control. The positive control was used just to compare and identify dead parasites. Even so, after your suggestion, we have reconsidered it and we mention the use of this positive control both in Material and Methods (lines 364-365) and in the Results (lines 145-146).

  1. The results may be compared with some known natural compounds or synthetic ones.

Your suggestion is certain and, although in our experiment, we did not compare with other compounds as this was not the aim of our study, in the discussion we mention the comparative results obtained by other authors using similar natural compounds such as carvacrol or thymol, and also synthetic ones (comparison with other antibacterial compounds in references 35, 36 and 37; comparison with other antiparasitic compounds in the references 61, 62,63 and 64).

Reviewer 2 Report

The manuscript of José F and colleagues present the antibacterial and antiparasitic activities of compounds PTS and PTSO through in vitro and in vivo studies. Firstly, the inhibitory activity of these two compounds was tested against four bacterial and parasitic species in vitro. Next, the effects of diets containing PTS and PTSO on gilthead sea bream juveniles infected with P. damselae subsp and on parasites/fish were evaluated by animal models.

The aim of this manuscript is clearly expressed and some literature in the field is applied. Yet, there are couple of suggestions which in my opinion should be addressed prior endorsing the manuscript for publication.

1. “In onion, the most common sulfur compounds are isoalliin (S-propenyl-L- cysteine sulfoxide); methiin (S-methyl-L-cysteine sulfoxide) and propiin (S-propyl-L- cysteine sulfoxide) that, due to the action of alliinase, leads to dipropyl thiosulfinate (PTS) which is transformed into dipropyl disulfide and propyl-propane thiosulfonate (PTSO) through dismutation or disproportionation reactions.” But how to generate the compound PTS  in onions? Can you show the content of compounds PTS and PTSO in onion? 

2. PTS and PTSO are the more stable structures among the onion organosulfur compounds, but have not been shown to be the best organosulfur compounds in terms of antibacterial and antiparasitic activity.

3. In the antibacterial part,  PTS and PTSO were tested for their inhibitory activity against several bacteria but no control was added (isoalliin or propiin or methiin).

4. In the experimental part,  the repair of the intestine by the compounds PTS and PTSO should be included.

5. In section 4.3., how to identify that the parasite is belonging to Sparicotyle chrysophrii.

6. What is the basis for setting the PTS/PTSO concentration at 150 mg/kg in the fish experimental diet part? What concentration of PTS/PTSO will be toxic to fish?

Author Response

The manuscript of José F and colleagues present the antibacterial and antiparasitic activities of compounds PTS and PTSO through in vitro and in vivo studies. Firstly, the inhibitory activity of these two compounds was tested against four bacterial and parasitic species in vitro. Next, the effects of diets containing PTS and PTSO on gilthead sea bream juveniles infected with P. damselae subsp and on parasites/fish were evaluated by animal models.

The aim of this manuscript is clearly expressed and some literature in the field is applied. Yet, there are couple of suggestions which in my opinion should be addressed prior endorsing the manuscript for publication.

  1. “In onion, the most common sulfur compounds are isoalliin (S-propenyl-L- cysteine sulfoxide); methiin (S-methyl-L-cysteine sulfoxide) and propiin (S-propyl-L- cysteine sulfoxide) that, due to the action of alliinase, leads to dipropyl thiosulfinate (PTS) which is transformed into dipropyl disulfide and propyl-propane thiosulfonate (PTSO) through dismutation or disproportionation reactions.” But how to generate the compound PTS in onions? Can you show the content of compounds PTS and PTSO in onion?

In the same way that happens with allicin in garlic when the alliin comes in contact with alliinase, the compound PTS (the saturated analogous of allicin) in onion is generated when, due to the action of crushing or cutting, the propiin, placed in the cytosol, comes in contact with the enzyme alliinase present in vacuoles, so the reaction takes place in the cytoplasm (Jones et al., 2004)

The content of the total OSCs in onion is 2-3% (Kim et al. 2016). From this total amount, the profile of these types of compounds, including PTS and PTSO is very difficult to estimate as it depends on the variety of the onion selected, the conditions under which they are cultivated, and the extraction process.

  • Jones, M. G., Hughes, J., Tregova, A., Milne, J., Tomsett, A. B., & Collin, H. A. (2004). Biosynthesis of the flavour precursors of onion and garlic. Journal of Experimental Botany, 55(404), 1903–1918. https://doi.org/10.1093/JXB/ERH138
  • Kim, S., Lee, S., Shin, D., & Yoo, M. (2016). Change in organosulfur compounds in onion (Allium cepa L.) during heat treatment. Food Science and Biotechnology, 25(1), 115. https://doi.org/10.1007/S10068-016-0017-7
  1. PTS and PTSO are the more stable structures among the onion organosulfur compounds, but have not been shown to be the best organosulfur compounds in terms of antibacterial and antiparasitic activity.

There are other organosulfur compounds derived from Allium with antimicrobial and antiparasitic activity, such as allicin or ajoene. However, these compounds are unique to garlic while PTS and PTSO are unique to onions. Unlike allicin, PTS and PTSO are more stable and more unknown in their potential functionalities and applications, hence the interest in studying their potential use in feed.

  1. In the antibacterial part, PTS and PTSO were tested for their inhibitory activity against several bacteria but no control was added (isoalliin or propiin or methiin).

Thank you for your suggestion, indeed in all our assays, as usually, we use a positive control. In this case, as we did not want to establish a comparison with other  Allium compounds or other products, it does not appear in the table for MBC, but the use of ceftazidime at 8 µg/mL has been considered (lines 351-352). Following your recommendations, we have included a sentence about the results of the positive control in the text (lines 136-137).

  1. In the experimental part, the repair of the intestine by the compounds PTS and PTSO should be included.

We greatly appreciate this suggestion, and it is true that in future experiments we will focus on how it affects at intestinal physiology level and we will also include changes in the microbiome. This paper is a first approach where the potential use of these molecules as antimicrobial and antiparasitic is revealed in fish and it is part of the activities under the framework of a research project in which there was no funding to cover studies on intestinal damage. Following your suggestion, in the manuscript, we have included a sentence indicating that further studies are needed to analyze its effect on the repair of intestinal damage (lines 251, 252).

  1. In section 4.3., how to identify that the parasite is belonging to Sparicotyle chrysophrii.

The donors were verified by specialists in marine parasitology, who previously identified the species of the parasite by light microscopy and subsequent verification by qPCR.

  1. What is the basis for setting the PTS/PTSO concentration at 150 mg/kg in the fish experimental diet part? What concentration of PTS/PTSO will be toxic to fish?

This concentration was determined based on the results obtained in vitro in the Rabelo-Ruiz study, 2022. In addition, in previous studies, we have carried out palatability tests, without observing adverse effects even at doses of 1000 ppm. Despite safety tests have been performed in mice indicating the absence of toxicity, in the future, we will carry out specific safety tests for fish. This paper is a first approximation to the use of these compounds in aquaculture, but it is true that there are still many aspects to be investigated, including safety.